# Adaptive Thin Film Temperature Sensor for Bearing’s Rolling Elements Temperature Measurement

**DOI:** 10.3390/s22082838

**Published:** 2022-04-07

**Authors:** Yunxian Cui, Pengfei Gao, Wuchu Tang, Guowei Mo, Junwei Yin

**Affiliations:** School of Mechanical Engineering, Dalian Jiaotong University, Dalian 116028, China; dlcyx007@126.com (Y.C.); gaopengfei198935@163.com (P.G.); tangwuchu@126.com (W.T.); moguowei00@163.com (G.M.)

**Keywords:** thin-film sensor, double row tapered roller bearing, roller temperature, self-adaptive

## Abstract

With the continuous improvement of train speeds, it is necessary to find the possible problems of bearings in time, otherwise they will cause serious consequences. Aiming at the characteristics of rapid temperature change of bearings, a thin film thermocouple temperature sensor was developed to measure the real-time temperature of the bearing’s rolling elements during train operation. Using dc pulse magnetron sputtering technology, Al_2_O_3_ film, NiCr film, NiSi film, and SiO_2_ film were successively deposited on an aluminum alloy substrate. We studied their microstructure, static characteristics, dynamic characteristics, and repeatability. Finally, we installed an adaptive film temperature sensor on the bearing testing machine to measure the temperature of the rolling elements. The results show that the developed temperature sensor has good linearity in the range of 30~180 ℃. The Seebeck coefficient is 40.69 μV/℃, the nonlinear fitting error is less than 0.29%, the maximum repeatability error is less than 4.55%, and the dynamic response time is 1.42 μs. The temperature of the measured rolling elements is 6~10 ℃ higher than that of the outer ring, which can reflect the actual temperature of the bearing operation.

## 1. Introduction

High-speed trains have become the most critical and advanced part of intelligent transportation, and attention has been paid widely to the requirements of high reliability and stability of trains to [1]. Bearing temperature monitoring of high-speed trains is one of the most important ways to ensure the safe operation of trains. If the axle box bearing of a high-speed train is abnormally hot, the warm or hot axle phenomenon appears, affecting the train’s regular operation. If severe, it cuts the shafts, seriously affecting railway transportation safety and even causing life and property losses [2]. Therefore, rapid and accurate measurement of the real-time temperature of high-speed train axle box bearings is of great significance to the safe operation of the train.

Currently, railway freight cars mainly use infrared temperature monitoring systems, and EMU mainly uses onboard platinum resistance to monitor the temperature. The former is a non-contact measurement, which is greatly disturbed by the environment. The latter has serious thermal hysteresis and cannot measure the rapidly changing shaft temperature [3]. In addition, the temperature measurement point of the axle box bearing in service is the outer surface of the outer ring, which is far away from the heat source and has a long temperature transfer path, which causes heat loss and noise interference. Therefore, the current railway bearing temperature monitoring system has sensor response problems and test location problems, which cannot accurately monitor the real-time temperature of the train running.

There have been few kinds of research on railway bearing temperature sensors in recent years, and the above problems have not been improved. In terms of sensor measuring points, Gao et al. [4,5] integrated the micro-sensor module into the raceway of the outer bearing ring. They proposed a new method for online real-time monitoring of rolling bearing running status. Shao et al. [6,7] proposed an intelligent bearing structure based on embedded multi-parameter sensors. In this structure, the two semi-ring supports and the outer bearing ring are slotted and fastened, and various sensors are embedded in the support ring to monitor the internal signals of the bearing. Insight technology proposed by SKF [8] integrates intelligent bearings, sensing rings composed of various sensors, spontaneous electrical devices, and wireless signal transmission structures. The sensing ring and bearing outer ring are welded by laser. At the same time, the Schaeffler group [9,10] also launched the FAG-Variosense series intelligent bearings that can be configured flexibly. Based on SKF Insight technology, the smart bearings modularized each function and assembled each module with a combined mounting shell for easy disassembly and replacement. Wang et al. [11] embedded the sensor temperature test system into the bearing lock nut and realized the contact temperature measurement through the contact between the lock nut and the bearing inner ring. These methods obtain more accurate signals related to bearings, but they destroy bearings to varying degrees, thus affecting the performance of bearings in all aspects.

Yan et al. [12,13,14] synthesized a quantum dot temperature-sensitive sensor based on quantum dot materials’ temperature sensing fluorescence characteristics to improve the sensor performance. They realized the temperature measurement of rotating parts under the service conditions of high-speed bearings. However, the synthesis process dramatically affects the sensor’s calibration and lifespan. The cost of the whole system is very high, and it is mainly used in the laboratory environment.

Thin-film thermocouple sensors have a fast response speed and small size. When the train runs at 360 km/h, a single roller through the sensor is 1.6 ms. For bearing roller temperature measurement, the response speed of the traditional thermocouple (10~30 s) is not sufficient. Due to the small internal space and complex environment of bearings, conventional thermocouples cannot be installed. Therefore, developing a thin-film thermocouple temperature sensor with a unique structure is necessary.

The axle box bearing of the train is a double row tapered roller bearing, whose rolling elements’ temperature is the highest [15]. At present, researchers have not reported on measuring temperature directly inside the bearing under the condition of ensuring the integrity of the main parts of the bearing. In this paper, an adaptive thin-film thermocouple sensor is designed and developed to measure the real-time temperature of the rolling elements of the train bearing. The sensor is fixed on the bearing seal cover and can accurately measure the rolling elements’ real-time temperature without damaging the bearing’s main structure. The sensor has the advantages of small size, fast response, good stability, and adaptive and wearable measuring end.

## 2. Working Principle and Structural Design

### 2.1. Working Principle

Employing the same working principle as ordinary thermocouples, the thermal electrodes of two different film materials of thin-film thermocouple sensor are connected end to end to form a closed loop. When there is a temperature difference between the contact points of the two thermal electrodes, the sensor loop will generate thermoelectric force, just as shown in Figure 1.

Peltier effect is the contact potential between two conductors.

Thomson effect is the temperature difference potential of a single conductor.
(1)$$S_{AB}=\frac{d\Pi }{dT}+\left(\mu_{B}-\mu_{A}\right)$$
where the first term is the Thomson effect, and the other represents the Peltier effect.

The Seebeck thermoelectric effect equals the algebraic sum of the Peltier thermoelectric effect and the Thomson thermoelectric effect in the circuit.

### 2.2. Structural Design

When the sensor measures the temperature of the rolling elements, it contacts and moves relative to the rolling elements. The self-adaptive thin-film thermocouple sensor developed in this paper adopts 6061 aluminum alloy to avoid damage to the rolling elements during temperature measurement because 6061 aluminum alloy has low hardness, good processing performance, high thermal conductivity, and a low thermal expansion coefficient and is compact and uniform, and the film is easy to coat as the base material. It is processed by ultra-precision wire cutting technology.

This paper takes the TAROL unit, tapered roller bearing used for high-speed trains, as the research object. The bearing has an outer diameter of 240 mm, an inner diameter of 130 mm, and a width of 160 mm, as shown in Figure 2a. Figure 2b shows the overall size of the sensor. The temperature measuring end is semicircular with a diameter of 5 mm, and the reference end is a cylinder with a diameter of 5 mm. The base structure of the temperature sensor is shown in Figure 3. One end of the base is a 1/4 sphere with a radius of 2.5 mm (measuring end), and the sensitive end is composed of Al_2_O_3_ insulating film, NiCr-NiSi functional film, and SiO_2_ protective film. The size of thermal contact is 0.5 × π × 2.5^2^ mm^2^. The other end is a semi-cylindrical surface with a radius of 2 mm (terminals). An M2.5 threaded hole is designed on the base to fix the base and groove with flat head set screws to ensure the reliable and firm connection between the compensation wire and the functional film; see Figure 4.

## 3. Development of Adaptive Thin-Film Thermocouple Temperature Sensor

### 3.1. Thin Film Temperature Sensor Function Film Preparation

The magnetron sputtering system used in this paper is the JGP450B multi-target magnetron sputtering system designed by the Shenyang Scientific Instrument Development Centre of the Chinese Academy of Sciences. The developed film thermocouple temperature sensor functional film mainly includes Al_2_O_3_ insulating film, NiCr functional film, NiSi functional film, and SiO_2_ protective film. Before preparing the functional film, the surface of the substrate should be ground and polished to a mirror surface. Then, the substrate, fixture, and mechanical mask should be placed in an ultrasonic cleaning machine, cleaned with acetone, alcohol, and deionized water, twice, respectively, 10 min each time, and dried with N_2_ air. The target material is cleaned by pre-sputtering to remove oxides and impurities on its surface. The substrate after cleaning is fixed on a particular chuck, and the special chuck is put into the vacuum magnetron sputtering equipment to prepare functional films. The sensor preparation process is shown in Figure 4.

#### 3.1.1. Preparation of Al_2_O_3_ Insulating Film

The aluminum alloy base has electrical conductivity. It is necessary to prepare insulating film on the surface of the base; otherwise, it will cause measurement errors due to the loss of thermoelectric force and even lead to abnormal measurements. Commonly used insulating films are Al_2_O_3_ film and SiO_2_ film. Al_2_O_3_ film can play the role of transition alloy bottom film because Al_2_O_3_ film is similar to aluminum alloy base material. In addition, it has high strength, stable chemical properties, corrosion resistance, wear resistance, good insulation, good thermal stability, and other advantages, so we choose Al_2_O_3_ film as the insulation film of the film sensor. The cleaned substrate is completely fixed on the special chuck. Then it is put into the vacuum chamber of the JZFZJ-500S high vacuum multifunctional composite coating machine. The DC pulse magnetron sputtering technology is used to deposit Al_2_O_3_ insulating film on the substrate. The sputtering deposition process parameters of Al_2_O_3_ insulating film are shown in Table 1.

The insulation resistance measured by DMM7510 Digital Multimeter is 2.6 × 10^8^ Ω, which is higher than the insulation resistance required by the national standard thermocouple (10^6^ Ω) and meets the insulation requirements of film temperature sensor used for temperature measurement of a rolling elements of high-speed train bearings.

#### 3.1.2. Preparation of NiCr/ NiSi Functional Thin Films

When measuring temperature with a thin-film thermocouple, there should be good insulation between the thermal electrodes, between the compensation wires, and between the thermocouple and the base; otherwise, it causes loss of thermoelectric force, resulting in errors [16]. Therefore, when preparing NiCr/NiSi functional films, masks need to be designed and installed, and the parts removed by covers are corresponding functional films. When coating sputtering, if the mask is too thick, it will block rays and form shadows, affecting the coating effect; if the mask does not fit well with the substrate, diffraction will occur, and even coating failure will occur. Therefore, the mask thickness is controlled within 0.5 mm. According to the above requirements, a multifunctional mask is designed. The installation method of the mask in the coating process is shown in Figure 5.

The base and the mask are entirely fixed on the particular chuck; NiCr functional film with a thickness of 800 nm is deposited on the surface of Al_2_O_3_ insulating film by the DC pulse magnetron sputtering technology. After preparing NiCr functional film, remove the multifunctional mask, turn it over, and install it. Replace NiSi functional film target and deposit NiSi functional film with a thickness of 800 nm. The overlap of NiCr/NiSi functional film forms the hot contact of the sensor (namely, the temperature measuring end), and the area of the hot contact is 0.5 × π × 2.5^2^ mm^2^. The preparation parameters of the NiCr/NiSi functional membrane are shown in Table 2.

#### 3.1.3. Preparation of SiO_2_ Protective Film

If NiCr and NiSi films are in contact with air for too long, the temperature measurement performance of films is affected. Moreover, the thin-film thermocouple temperature sensor developed in this paper moves relative to the rolling elements of the bearing, resulting in wear. SiO_2_ protective film should be prepared on the surface of the functional film to play a role in oxidation resistance and wear resistance to prolong the sensor’s service life. Flip the multifunctional mask and install it again, as shown in Figure 5d. A SiO_2_ protective film with a thickness of 2000 nm is deposited on the surface of the functional film. The film deposition process parameters are shown in Table 3.

### 3.2. Lead-Film Mechanical Clamping Construction

The lead wire is the channel of electrical signal transmission of the thin-film temperature sensor. Suppose the lead is in poor contact with the film or the lead falls off. In that case, the collected signal data does not accurately reflect the current temperature and the characteristics of the film thermocouple.

The film and the lead are mainly connected by conductive silver glue. The conductive silver adhesive falls off easily, reducing the strength of the connection between the lead and the film sensor [17]. Therefore, a lead-film mechanical clamping structure is proposed in this paper. As shown in Figure 4, 2.6 μm Al_2_O_3_ insulating film was deposited on the surface of the groove structure. NiCr/NiSi functional film deposited on aluminum alloy substrate corresponds to NiCr/NiSi compensation wire; buckle the prepared base and groove structure, and then, tighten the flat end set screw to tighten the compensation wire. 

### 3.3. Design of Sensor Adaptive Function

A high-speed train bearing is a double row tapered roller bearing. The friction between the cone surface of the rolling elements and the raceway of the inner and outer ring produces the most heat. The heat production efficiency of the big end of the rolling elements and the big rim of the inner ring is the highest per unit time. Because of the minimum radius of curvature, the temperature of the rolling elements is the highest. The large end face of the roller is a spherical structure, and the adjacent rollers are discontinuous.

Figure 6 shows the physical picture of the adaptive thin-film temperature sensor. The temperature sensor realizes the adaptive function through a sleeve, spring, base, and back cover. The casing and base structure realize the sensor’s guidance, and the length of the casing controls the expansion of the sensor. The sensor is fixed on the bearing seal ring with a set screw. The hemispherical structure of the measuring end of the sensor is in contact with the spherical surface of the big end of the roller to achieve adaptive measurement and wear.

Finally, the gaps at both ends of the assembled thermocouple sensor are filled with high temperature resistant thermal conductive adhesive to avoid water, grease, abrasive, and other impurities into the thermocouple.

## 4. Results and Discussion

### 4.1. Surface Morphology Analysis of Thin Films

During the preparation of each film, several slides were placed as samples to prepare the same furnace. The surface morphology of samples prepared in the same furnace for each thin film layer was observed by JEM-2100F Scanning Electron Microscope (SEM). See Figure 7.

Figure 7 shows the SEM images of Al_2_O_3_, NiCr, NiSi, and SiO_2_ thin films. It shows that the surface of Al_2_O_3_, NiCr, NiSi, and SiO_2_ films are flat, dense, evenly distributed, and free of apparent defects, which meets the requirements of the film thermocouple temperature sensor to prepare.

### 4.2. Composition Analysis of Thin Films

We analyzed the composition of deposited Al_2_O_3_, NiCr, NiSi, and SiO_2_ films by the Energy Dispersive Spectrometer (EDS). The results are shown in Figure 8.

Figure 8 shows that The O and Al atom composition ratio of Al_2_O_3_ film is 64:36, close to the target atom ratio of 3:2. The Ni and Cr atom composition ratio of NiCr film is 90.81:9.19, close to the target atom ratio of 90:10. The Ni and Si atom composition ratio of NiSi film is 97.02:2.98, close to the target atom ratio of 97:3. The Si and O atom composition ratio of SiO_2_ thin film is 34.45:65.55, close to the 1:2 atomic ratio of SiO_2_. The experimental analysis results show that the composition ratio of Al_2_O_3_, NiCr, NiSi, and SiO_2_ films evidently meets the requirements.

### 4.3. Surface Morphology and Roughness Analysis 

We observed the microscopic surface morphology and surface roughness of samples prepared in the same furnace for each thin film layer using the Multimode8 Atomic Force Microscope (AFM). The results are shown in Figure 9. The roughness of Al_2_O_3_, NiCr, NiSi, and SiO_2_ films is 1.3 nm, 5.8 nm, 8.5 nm, and 11.5 nm, respectively. The four films have good uniformity, continuity, and densification on the whole.

### 4.4. Static Characteristic of Thin Film Thermocouple Temperature Sensor

Sensor calibration is essential in designing, manufacturing, and using sensors [18]. Seebeck coefficient and response time are static and dynamic indexes of thin-film thermocouple temperature sensors [19]. They must be calibrated after the sensor is manufactured and packaged to ensure that the sensor accurately measures the transient temperature change.

#### 4.4.1. Static Calibration 

The static calibration system of the thin-film thermocouple temperature sensor is composed of FLUKE-9144 Dry Metering Furnace, FLUKE-9170 Dry Metering Furnace, and Tektronix-DMM7510 Digital Multimeter. The static calibration system is shown in Figure 10.

When the absolute bearing temperature is more significant than 120 ℃, the bearing monitoring device sends out a hot shaft pre-alarm signal; when the absolute bearing temperature is more excellent than 140 ℃, the bearing monitoring device sends a hot shaft alarm signal. When the vehicle gives an alarm signal, the speed limit of EMU is 220 km/h. If the shaft temperature continues to rise, the vehicle automatically brakes and stops [20]. This experiment’s static calibration temperature range is set to 30~180 ℃ because the FLUKE-9114 dry metering furnace temperature range is 30~660 ℃ and has a maximum of 8 preset temperature points. It ensures that the thin-film sensor has a high measurement accuracy in regular operation and can still normally measure the shaft temperature in the case of the hot shaft.

We placed the hot end of the developed film thermocouple temperature sensor in the FLUKE-9144 dry metering furnace. We filled high-temperature glass wool with the sensor to insulate from the outside. We set FLUKE-9170 dry metering furnace to 0 ℃ constant used as freezing point thermostat, and compensation wire (i.e., cold end) was placed in it. We set the static calibration temperature range from 30 ℃ to 180 ℃ to adjust the FluKE-9144 dry measuring furnace. We kept the temperature for 5 min every time the temperature increased by 10 ℃, used a digital multimeter to record the thermoelectric force value changing at different temperatures, and obtained the relationship between the thermoelectric force of the film thermocouple temperature sensor and the static calibration temperature, as shown in Table 4.

The least-square method is used to fit the static calibration data linearly. The static calibration curve, fitting equation, Seebeck coefficient, and linear correlation suitable coefficient of the temperature sensor are obtained. The proper curve and equation are shown in Figure 11.

Where E and θ are output thermoelectric force and temperature, respectively. 

The results show that the Seebeck coefficients (sensitivity) of the prepared thin-film thermocouple temperature sensors are 40.76 μV/℃ and 40.69 μV/℃, respectively; the maximum nonlinear errors are less than 0.29%, and the linear correlation coefficients are above 0.999.

#### 4.4.2. Repetitive Experiment

The same thin-film thermocouple temperature sensor was calibrated six times using the above test method to ensure that the developed sensor has good repeatability. The data fitting results are shown in Figure 12.

We can see from Figure 12 that the six calibration results of the adaptive thin-film thermocouple temperature sensor are similar, and the repeatability error analysis of the data results is carried out:(2)$$\sigma =\sqrt{\frac{\sum {\left(x_{i}-x\right)}^{2}}{n-1}}$$
where *n* is the actual number of measurements, *x_i_* is the result of each test, *x* is the average of multiple measures, and *σ* represents the standard deviation.
(3)$$\delta =\frac{\sigma }{x}\times 100\%=4.55\%$$

The repeatability experiment results show the sensor’s maximum repeatability error is 4.55%. The data show that the developed thin-film thermocouple temperature sensor has good repeatability and can measure temperature repeatedly and accurately.

### 4.5. Dynamic Performance Test of Thin Film Thermocouple Temperature Sensor

In this paper, a high-speed train TAROL bearing unit is selected, the number of rolling bodies is 17, and the average diameter is 27 mm. The train running speed is 360 km/h, and the wheel diameter is 860 mm. By calculation, the corresponding wheel speed is 2220 r/min, and the time for a single rolling element to pass a fixed point is 1.6 ms.

The dynamic response time of the sensor is measured by the pulse response method. The dynamic calibration system includes Quantel Laser Ultra 50 short-pulse laser and DMM7510 Digital Multimeter. The dynamic test system is shown in Figure 13.

We connect the compensation wire of the film thermocouple temperature sensor to the DMM7510 Digital Multimeter and set the sampling rate, storage, and other parameters. The temperature sensor is fixed to the target frame of the test stand. The red-light source of the HE-Ne laser is adjusted to focus on the hot contact position of the thin-film thermocouple temperature sensor. We change the laser source to correspond to the laser red light source and then adjust the laser voltage to 640 V and the laser energy to 18.25 MJ and choose the automatic emission mode to the output laser pulse. Finally, the DMM7510 Digital Multimeter displays and stores the dynamic calibration curve in real-time. The pulse response curve of the thermocouple temperature sensor is shown in Figure 14.

According to the curve calculation, the response time of the thin-film thermocouple temperature sensor is 1.42 μs, and the response time is a microsecond. Therefore, the sensor can meet the requirements (1.6 ms) of temperature measurement of the bearing’s rolling elements during train operation.

### 4.6. Temperature Measurement Experiment of Thin Film Thermocouple Temperature Sensor Bearing

The outer bearing ring is fixed and clamped on the test table, and the sealing cover is fitted with the bearing exterior ring interference. The thin-film thermocouple temperature sensor is specified on the seal cover by a set screw, and the temperature signal is transmitted to the Digital Multimeter by wired mode. Figure 15 shows a railway bearing temperature test device.

The bearing temperature experiment was carried out on the bearing dynamic quality testing machine to explore the practicability of the thin-film thermocouple temperature sensor in the process of bearing operation. The high-speed train TAROL unit and bearing of wagon 353130B are double-row tapered roller bearings with similar structures. This paper focuses on improving the hysteresis of the train’s temperature measurement system, focusing on improving the response speed of the temperature sensor and the rationality of the layout of temperature measurement points. Therefore, selecting a 353130B as the experimental object does not affect the testing purpose. The infrared temperature measuring gun is fixed on the test table to calculate the bearing outer ring temperature (the current temperature measuring point of high-speed train axle box bearing), as shown in Figure 15.

As shown in Figure 16, the spindle runs at a speed of 400 r/min, and the temperature of the bearing’s rolling elements begins to rise. After about 1 min, the temperature of the outer ring increases significantly, the temperature rise rate of the outer bearing ring is always less than that of the rolling elements. When the external ring temperature reaches 45 ℃, the lag time is about 8 min. After 15 min, the bearing temperature tends to be stable; the stable temperature of the rolling elements is 55.8 ℃; the stable temperature of the outer ring is 46.4 ℃, and the temperature difference is 9.4 ℃. 

The heat is generated by the sliding friction between the rolling elements and the inner and outer raceway. On the other hand, it is caused by the sliding friction between the rolling elements and the inner ring big rim. Heat is then transferred to the outer surface of the outer ring by heat conduction. The stable temperature measured by the thin-film thermocouple sensor is higher than that measured by the infrared temperature gun because of the heat loss during heat transfer. The temperature measured by the infrared temperature gun cannot reach the stable temperature measured by the film thermocouple sensor. That is, the outer surface temperature of the outer ring cannot reflect the actual state of the bearing.

Under laboratory conditions (natural convection of air), the spindle speed is 400 r/min, 600 r/min, 800 r/min, and 1000 r/min, respectively. When the bearing temperature is stable, the temperature measured by the film thermocouple temperature sensor and infrared temperature gun is shown in Table 5.

As shown in Table 5, when the spindle speed is 1000 r/min, the stable temperature exceeds 110 ℃ (the hot shaft warning temperature is 120 ℃), so it is set as the maximum design experimental speed. The bearing testing machine runs at a spindle speed of 600 r/min for 45 min, then runs at a spindle speed of 200 r/min for 45 min, and then stops. To simulate the train running process, the train in the running process includes starting, accelerating, running at the full design speed, then slowing down, running at a consistent pace, slowing down again, and stopping.

As shown in Figure 17, temperature lag time is a variable that tends to decrease gradually as the temperature increases. When the temperature of the surrounding medium rises, the heat loss generated in the heat transfer process decreases, and the temperature lag decreases. The curve measured by the film sensor fluctuates more wildly (many burrs). When we use the film sensor to measure the roller’s temperature, there is a very short interval between adjacent two rollers in the process of bearing rotation. The roller’s temperature and the oil and gas mixture temperature are slightly different. Hence burr belongs to the normal phenomenon; simultaneously, this sensor response speed can reflect the rolling elements temperature.

We found the sensor’s wear during our experiment, but it is tiny. Because an oil film is formed between the contact surfaces during the bearing’s working process, it dramatically reduces friction heat generation. When the roller moves to the center of the sensor (the spring is compressed to the shortest amount), the design elastic force of the spring is 1.4 N, and the sensor and the roller are in point contact, so the friction generated is minimal, and the friction heat is very little. The influence on the bearing’s dynamics is also minimal. The sensor’s base material is aluminum alloy, which will not affect the roller even if it is slightly worn. While the sensor’s measuring end is a self-renewal structure, a small amount of wear will not affect the sensor’s performance.

Additionally, we can see a temperature difference in Figure 16 and Figure 17 because the sensor is in contact with the roller to measure the temperature. The temperature of the infrared measuring point is obtained through heat transfer, which will produce heat loss. We conducted a large number of bearing temperature measurement tests with sensors. Sensor failure only occurred once, caused by the breakage of the compensation wire, and there was no problem with the sensor itself. The result shows that the sensor is reliable.

## 5. Conclusions

(1) Using Magnetron sputtering technology to deposit Al_2_O_3_ insulating film, NiCr functional film, NiSi functional film, and SiO_2_ protective film on 6061 aluminum alloy substrate successively, allows to realize leader-film mechanical clamping and solve the problem presented by the conductive silver adhesive connection easily falling off.

(2) The microscopic morphology of the sensor film was characterized, and the results met the requirements. The static and dynamic calibration of the developed thin-film thermocouple temperature sensor shows that the sensor’s sensitivity is 40.69 μV/℃; the maximum repeatability error is 4.55%, and the dynamic response time is 1.42 μs.

(3) The developed film thermocouple temperature sensor is applied to bearings’ dynamic quality testing machine. The results show that the temperature measured by the rolling elements is 6~10 ℃ higher than the outer ring temperature, and the temperature change curve can reflect the actual state of bearings.

However, much remains to be done, such as high-speed railway bearing experiments, actual locomotive experiments, long-term data monitoring of sensors, and so on. By constantly improving the performance of sensors, new ideas are provided for the research of intelligent bearings, and reference is provided for enhancing the temperature monitoring system of a train bearing to provide a safety guarantee for train speed increase.

## Figures and Tables

**Figure 1 sensors-22-02838-f001:**
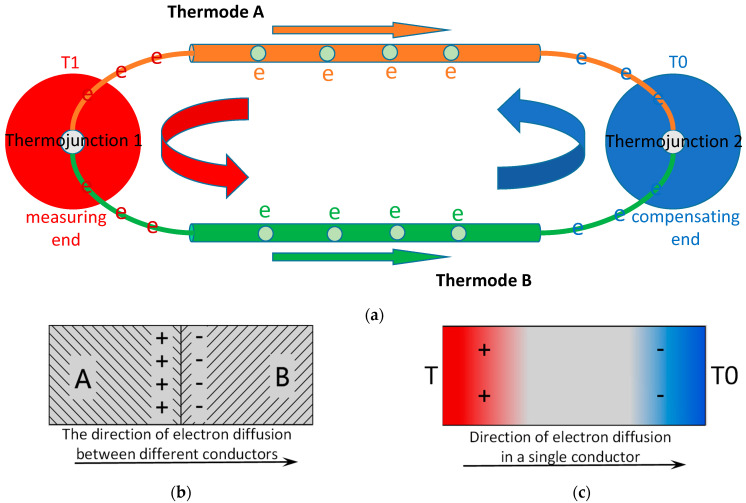
Working principle of thin-film thermocouple sensor: (**a**) Seebeck thermoelectric effect; (**b**) Peltier effect; (**c**) Thomson effect.

**Figure 2 sensors-22-02838-f002:**
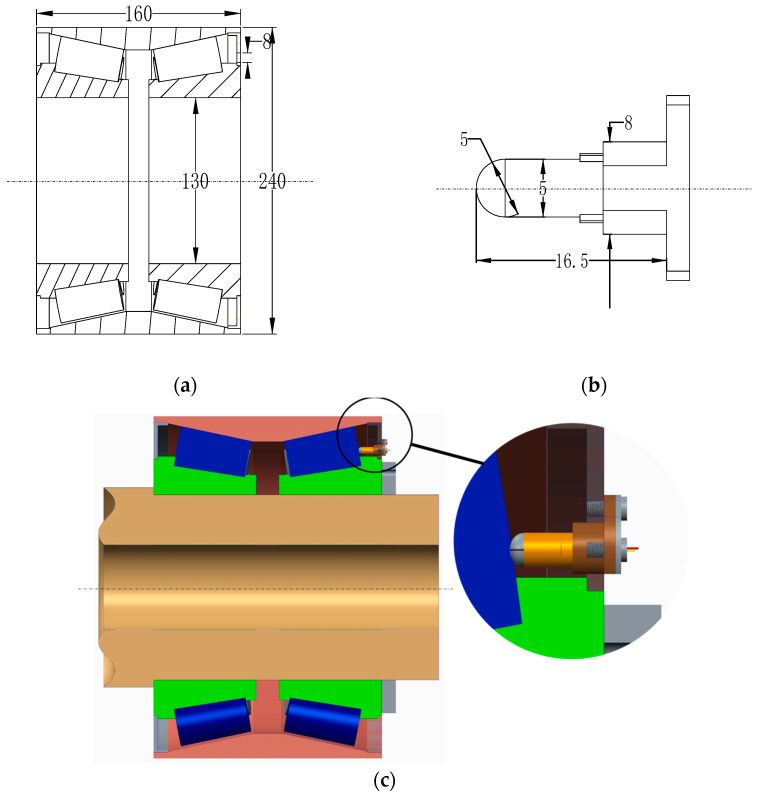
Bearing-sensor structure and assembly diagram: (**a**) TAROL unit structure; (**b**) temperature sensor structure; (**c**) bearing-sensor assembly diagram.

**Figure 3 sensors-22-02838-f003:**
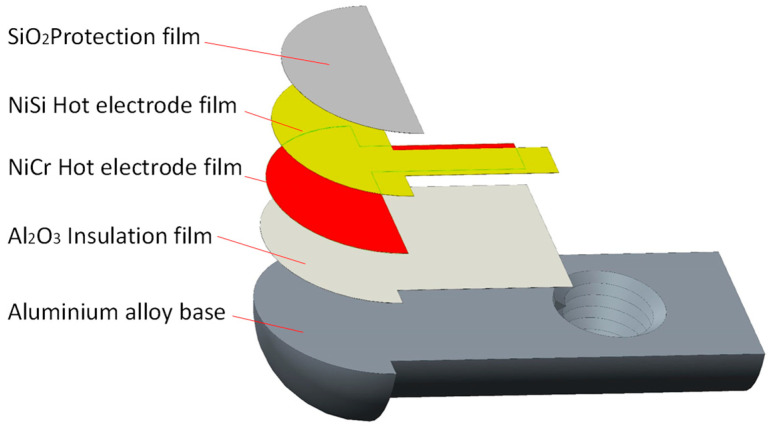
Schematic diagram of sensor base structure.

**Figure 4 sensors-22-02838-f004:**
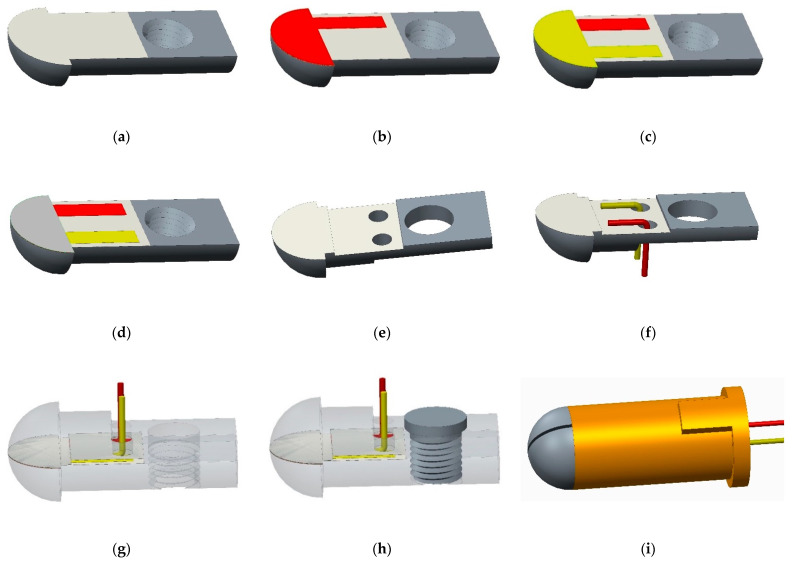
The preparation and assembly process of adaptive film thermocouple temperature sensor: (**a**) deposition of Al_2_O_3_ insulating film; (**b**) deposition of NiCr functional films; (**c**) deposition of NiSi functional films; (**d**) deposition of SiO_2_ protective film; (**e**) deposition of Al_2_O_3_ insulating film with groove structure; (**f**) compensation conductor passes through the through-hole of the groove structure; (**g**) base and groove structure snap together; (**h**) flat head screws secure the base and recessed structure; (**i**) hot pressing assembly sheathed sleeve.

**Figure 5 sensors-22-02838-f005:**
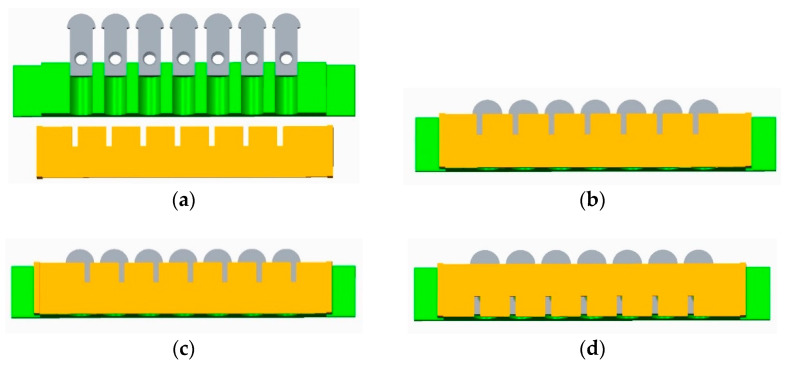
Installation method of the mask during coating process: (**a**) jig-base-mask installation diagram; (**b**) deposition NiCr functional film assembly drawing; (**c**) assembly drawing of deposited NiSi functional film; (**d**) assembly drawing of deposited SiO_2_ protective film.

**Figure 6 sensors-22-02838-f006:**
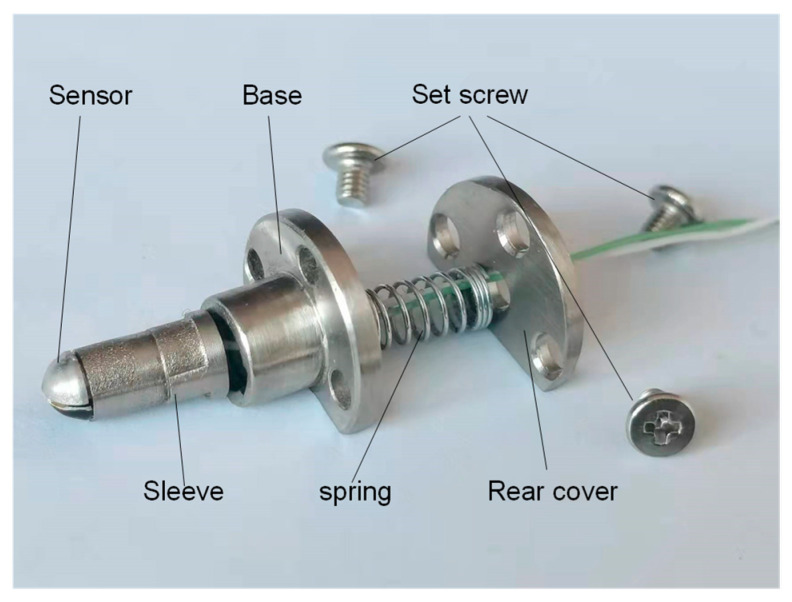
Actual picture of the adaptive film temperature sensor.

**Figure 7 sensors-22-02838-f007:**
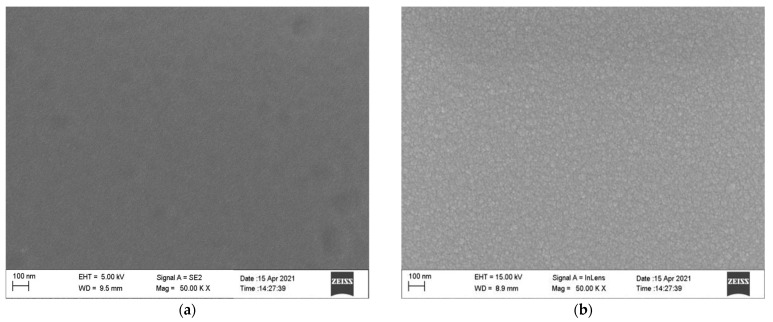
SEM image of functional film surface: (**a**) Al_2_O_3_ thin films surface; (**b**) NiCr thin films surface; (**c**) NiSi thin films surface; (**d**) SiO_2_ thin films surface.

**Figure 8 sensors-22-02838-f008:**
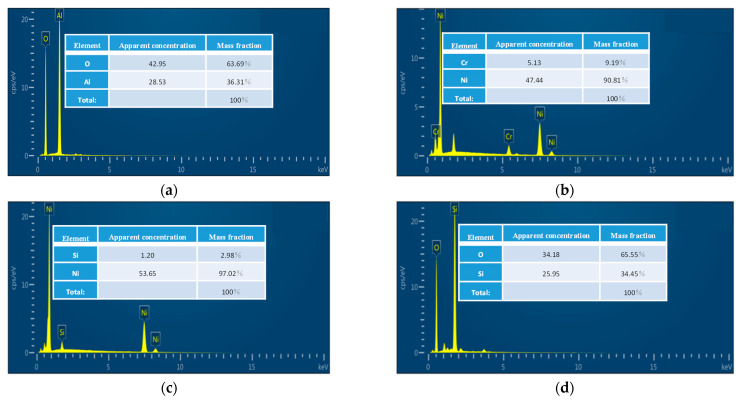
Energy spectrum analysis of thin films: (**a**) Al_2_O_3_ thin films; (**b**) NiCr thin films; (**c**) NiSi thin films; (**d**) SiO_2_ thin films.

**Figure 9 sensors-22-02838-f009:**
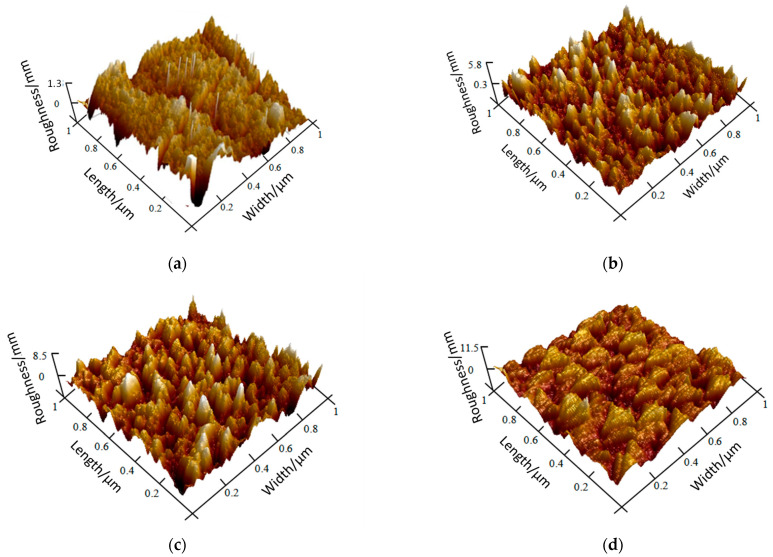
AFM diagram of functional thin films: (**a**) Al_2_O_3_ functional thin films, (**b**) NiCr functional thin films, (**c**) NiSi functional thin films, (**d**) SiO_2_ functional thin films.

**Figure 10 sensors-22-02838-f010:**
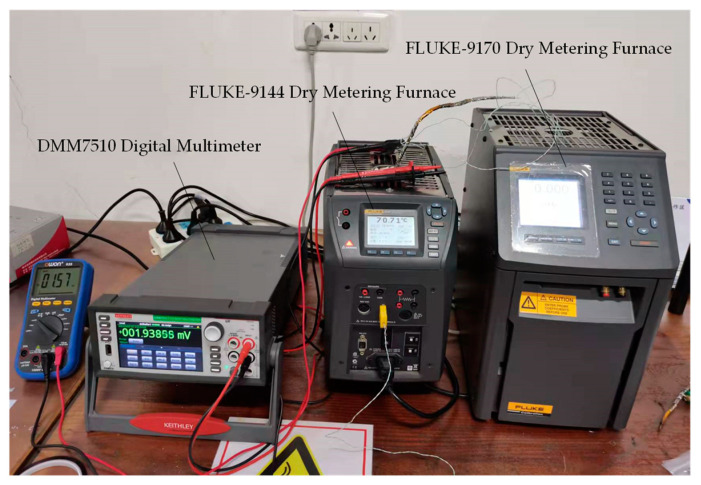
A thin film thermocouple temperature sensor static calibration system.

**Figure 11 sensors-22-02838-f011:**
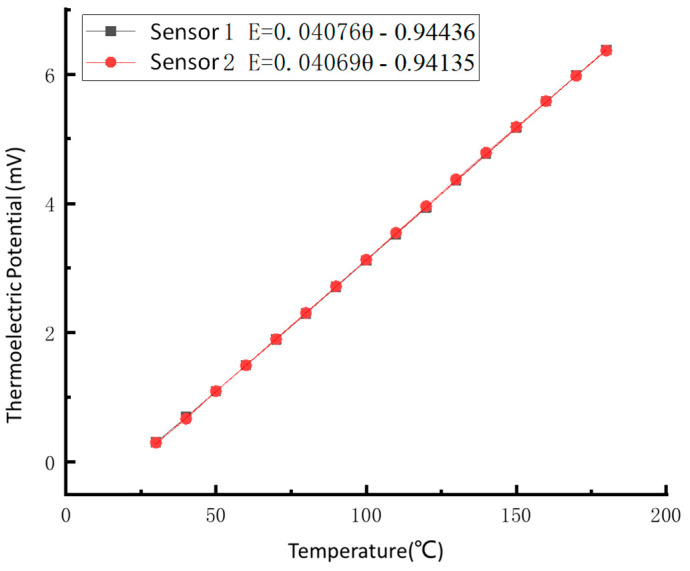
Static calibration curve of thin-film thermocouple temperature sensors.

**Figure 12 sensors-22-02838-f012:**
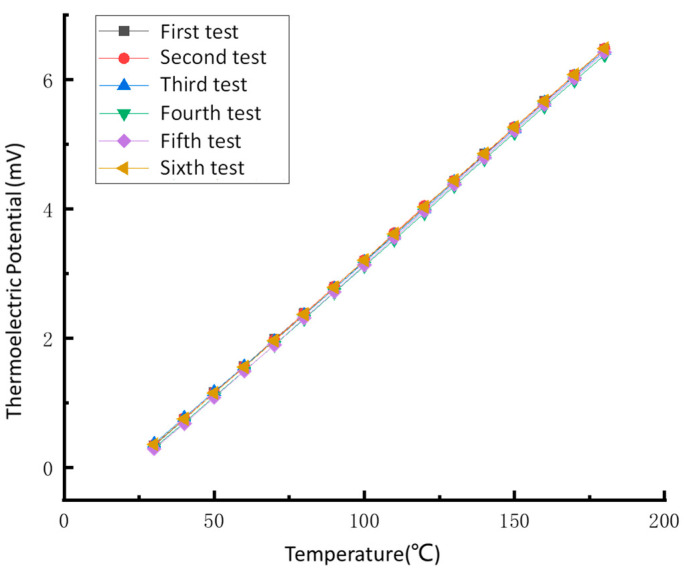
Repeatability calibration curve of thin-film thermocouple temperature sensors.

**Figure 13 sensors-22-02838-f013:**
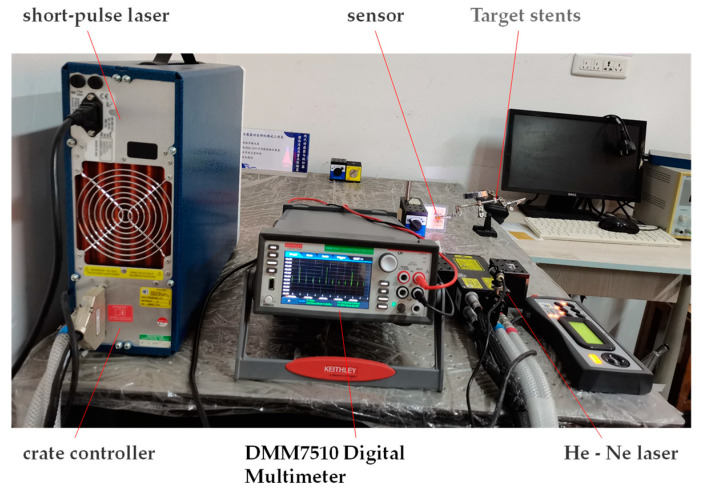
A thin film thermocouple temperature sensor dynamic calibration system.

**Figure 14 sensors-22-02838-f014:**
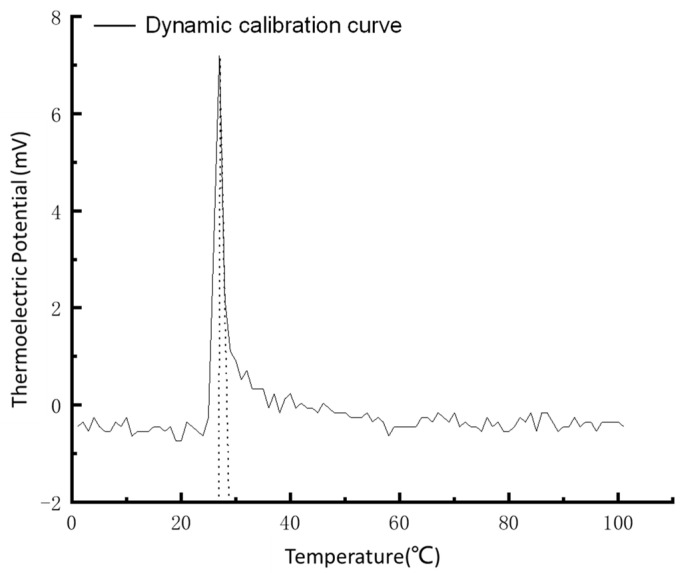
Dynamic calibration curve of a thin-film thermocouple temperature sensor.

**Figure 15 sensors-22-02838-f015:**
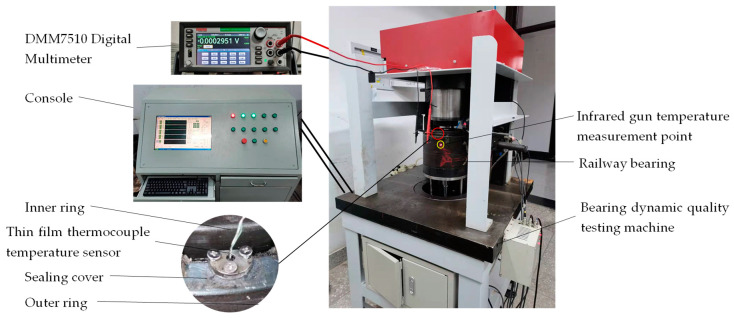
Railway bearing temperature measuring device.

**Figure 16 sensors-22-02838-f016:**
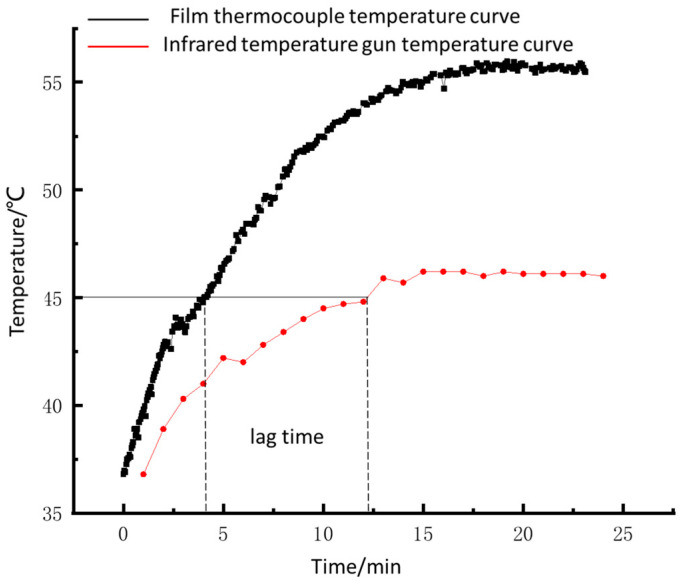
Spindle speed 400 r/min shaft temperature curve.

**Figure 17 sensors-22-02838-f017:**
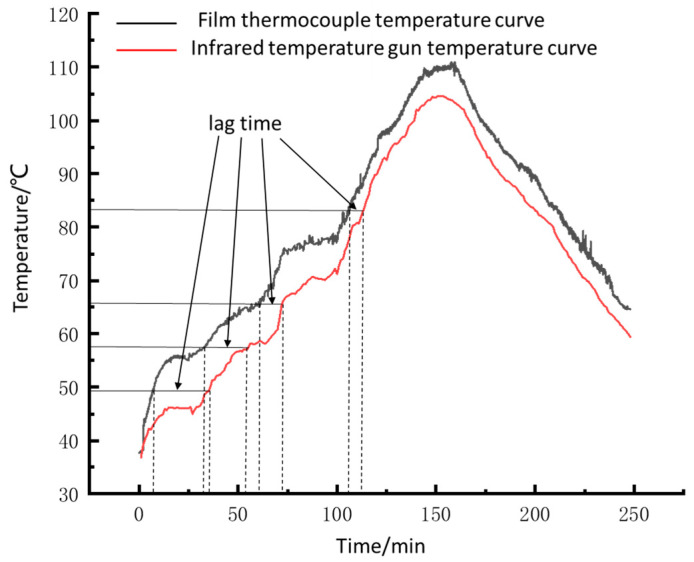
The dynamic curve of bearing temperature.

**Table 1 sensors-22-02838-t001:** Preparation process parameters of Al_2_O_3_ insulating film.

Feature Film	Background Vacuum (Pa)	Working Pressure (Pa)	Ar Flow (sccm)	O_2_ Flow (sccm)	Power (W)	Sputtering Time (H)
Al_2_O_3_	6.0 × 10^−3^	0.6	20	4	1000	3

**Table 2 sensors-22-02838-t002:** Preparation technological parameters of NiCr/NiSi thin film.

Feature Film	Background Vacuum (Pa)	Working Pressure (Pa)	Ar Flow (sccm)	Power (W)	Sputtering Time (min)
NiCr	7.0 × 10^−3^	0.7	20	350	40
NiSi	7.0 × 10^−3^	0.7	20	350	39

**Table 3 sensors-22-02838-t003:** Preparation process parameters of SiO_2_ protective film.

Feature Film	Background Vacuum (Pa)	Working Pressure (Pa)	Ar Flow (sccm)	O_2_ Flow (sccm)	Power (W)	Sputtering Time (min)
SiO_2_	6.0 × 10^−3^	0.6	20	10	600	120

**Table 4 sensors-22-02838-t004:** Relationship between thermoelectric force E and the temperature of the measuring terminal T.

Temperature Value (℃)	Sensor 1 Thermoelectric Force(mV)	Sensor 2 Thermoelectric Force(mV)
30	0.297	0.305
40	0.669	0.696
50	1.098	1.095
60	1.493	1.495
70	1.897	1.894
80	2.306	2.299
90	2.719	2.710
100	3.129	3.115
110	3.544	3.525
120	3.958	3.935
130	4.372	4.354
140	4.785	4.766
150	5.183	5.176
160	5.583	5.583
170	5.973	5.980
180	6.364	6.376

**Table 5 sensors-22-02838-t005:** Temperature contrast diagram of film thermocouple temperature sensor and infrared temperature gun.

Spindle Speed (r/min)	400	600	800	1000
temperature stabilization time (min)	20	40	20	65
sensor stabilization temperature (℃)	55.8	66.8	77.6	110.3
Infrared stabilization temperature (℃)	46.4	58.2	70.5	103.8
Temperature difference (℃)	9.4	8.6	7.1	6.5

## Data Availability

Data in this paper are available from the corresponding authors upon request.

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
