# Peer review of "Adaptive Thin Film Temperature Sensor for Bearing’s Rolling Elements Temperature Measurement"

_sensors, 2022, doi:10.3390/s22082838_

Round 1
Reviewer 1 Report
The manuscript introduces adaptive thin film temperature sensors for rolling bearings in the context of high speed trains. Thereby, Al2O3, NiCr, NiSi, and SiO2 films were deposited on aluminium alloy by means of dc pulse magnetron sputtering. The authors analyzed the microstructure, static and dynamic characteristics and repeatability. Finally, the delevoped sensor was installed on a rolling bearing test-rig to measure the rolling element temperature, providing plausible restuls.
In general, the topic is highly relevant in the context of condition monitoring of rolling bearings as well as the development toward sensor-carrying machine elements. Therefore, the study is of interest for the research community and industry. Basically, the manuscript is well written and illustrated, the methods and results are scientfically sound. However, prior to acceptance, I have a couple of comments that should be addressed:
1. My biggest concern is that a limiting factor of the chosen approach is the contact between the temperature sensor and the rolling element. Also, to my understanding, the sensor does not follow the roller element since it is fixed to the ring while the set of rollers is moving. So the sensor element keeps getting in touch with on roller, loosing contact and gets in contact with the next roller again. Correct? Anyways, the sliding and impact contact leads to wear on both the sensor and the rolling element. In addition, this possibly influences the friction losses of the entire bearing, and potentially also the dynamics. Also, I would expect frictional heating in the sliding contact between the roller and the sensor (and the loss of contact between the rollers) to influence the measured temperature. Might this explain the higher measured temperature compared to the infrared method in Fig. 16 and 17? How to ensure long-term stability of the sensor approach? The authors should at least discuss on these aspects.
Furthermore, I have a couple of minor comments:
2. "Bearing Rolling" should be changed to "Rolling Bearing" in the title.
3. Might be better to speak of "roller elements" instead of "roller body" throughout the manuscript.
4. When citing works from other authors, it is enough to provide the last names, e.g. "Gao et al. [4,5]" instead of "Robert X. Gao et al.".
5. Maybe the authors could differentiate their approach, which is a sensor-carrying machine element, from sensor-integrating and sensory utilizable machine elements (see https://link.springer.com/article/10.1007/s10010-019-00382-1) either in the introduction?
6. Fig. 2: Please at symmetry lines to your drawings and the Cad image.
7. The methods and equipment used for characterization of the sensors should be described in the methods section and not in the results. Section 4 should contain the results and discussion only.
8. Fig. 8 is hard to read and should be improved.
9. Table 5 is a bit confusing, the headings on the rows are not clearly related to the values (stabulization temp is next to sensor temp).
10. Line 383-383: Improve the wording (there's "on the other hand" twice).
Reviewer 2 Report
The test performance of multilayer thin film based thermocouple temperature sensor for conditions of rolling friction is discussed. Structural design of pin shaped sensor, preparation of insulating and functional films and their morphology is discussed. Sensor calibration showed almost linear dependence between thermoelectric potential and temperature for 30-180 degrees Celsius. The difference between measured temperatures by proposed sensor and infrared gun was about 9 degrees which justifies the choice of the sensor in applications.
Some possible applications of the discussed sensor in wear-fatigue tests including the ones for new structural materials of wheel/rail system could be referred to in the paper:
(i) a method of experimental study of friction in a active system, (ii) state of volumetric damage of tribo-fatigue system, (iii) spatial stress-strain state of tribofatigue system in roll-shaft contact zone, (iv) modeling of the damaged state by the finite-element method on simultaneous action of contact and noncontact loads, (v) tribo-fatigue behavior of austempered ductile iron monica as new structural material for rail-wheel system, (vi) research on tensile behaviour of new structural material monica, (vii) measurement and real time analysis of local damage in wear-and-fatigue tests
Variables in (1) should be defined.
The paper “Adaptive Thin Film Temperature Sensor for Bearing Rolling Body Temperature Measurement” could be published in Sensors after minor revision.
Reviewer 3 Report
Dear Authors,
Figure 7 depicts SEM images of thin-film Al2O3, NiCr, NiSi, and SiO2. However, there is no proper justification for every thin film, and it is also not mentioned as labelling. Therefore, the exact mechanism of thin-film formation is not discussed at all.
Figure 9: AFM diagram of a functional thin film is not precisely mentioned in the diagram, and the working mechanism is not focused.
Figure 11 shows which sensors are used and how they are interconnected to measure thin-film thermal behaviour.
Figure 14 shows a sudden increase in dynamic calibration at a specific temperature.
Why? Explain the mechanism's nature in detail.
The railway bearing temperature measuring device is not properly optimised and does not provide a valid justification for obtaining the best value.
Round 2
Reviewer 1 Report
Thank you for answering my comments. Unfortunately, some of them were not implemented in the manuscript. Before acceptance, I would like to see the answers and clarifications at least to my first comment also beeing actually reflected in the methods and/or discussion section of the paper so that other readers can profit from it.
Author Response
Your preciseness is really respected. I have sorted my answers and clarifications into the manuscript. Thank you very much!